# Contemporary patterns in kidney graft survival from donors after circulatory death in the United States

**Catherine R. Butler**[1], **James D. Perkins**[2], **Christopher K. Johnson**[3], **Christopher D. Blosser**[1], **Iris De Castro**[1], **Nicolae Leca**[1], **Lena Sibulesky**[2]*

1 Division of Nephrology, Department of Medicine, University of Washington, Seattle, WA, United States of America, 2 Division of Transplant Surgery, Department of Surgery, University of Washington, Seattle, WA, United States of America, 3 Division of Nephrology, Department of Medicine, University of Illinois College of Medicine, Peoria, IL, United States of America

* lenasi@uw.edu

## Abstract

**Data Availability Statement:** The data underlying the results presented in this study are available by request from the Scientific Registry of Transplant Recipients at: https://www.srtr.org/requesting-srtr-

### Background

Kidney transplants from donors after circulatory death (DCD) make up an increasing proportion of all deceased donor kidney transplants in the United States (US). However, DCD grafts are considered to be of lower quality than kidneys from donors after brain death (DBD). It is unclear whether graft survival is different for these two types of donor kidneys.

### Materials and methods

We conducted a retrospective cohort study of US deceased donor kidney recipients using data from the United Network of Organ Sharing from 12/4/2014 to 6/30/2018. We employed a Cox proportional hazard model with mixed effects to compare all-cause graft loss and death-censored graft loss for DCD versus DBD deceased donor kidney transplant recipients. We used transplant center as the random effects term to account for cluster-specific random effects. In the multivariable analysis, we adjusted for recipient characteristics, donor factors, and transplant logistics.

### Results

Our cohort included 27,494 DBD and 7,770 DCD graft recipients transplanted from 2014 to 2018 who were followed over a median of 1.92 years (IQR 1.08–2.83). For DCD compared with DBD recipients, we did not find a significant difference in all-cause graft loss (hazard ratio [HR] 0.96, 95% confidence interval [CI] 0.87–1.05 in univariable and HR 1.03 [95% CI 0.95–1.13] in multivariable analysis) or for death-censored graft loss (HR 0.97 (95% CI 0.91–1.06) in univariable and 1.05 (95% CI 0.99–1.11) in multivariable analysis).

### Conclusions

For a contemporary cohort of deceased donor kidney transplant recipients, we did not find a difference in the likelihood of graft loss for DCD compared with DBD grafts. These findings

data/data-requests/ The Scientific Registry of
Transplant Recipients is a 3rd party. The authors
do not have special privileges so others are able to
access the data in the same manner.

**Funding:** Dr. Butler is supported by a training grant
from the NIDDK (5T32DK007467-33).

**Competing interests:** The authors have declared
that no competing interests exist.

signal a need for additional investigation into whether DCD status independently contributes
to other important outcomes for current kidney transplant recipients and indices of graft
quality.

## Introduction

In 2016, nearly 100,000 patients were listed on the United States (US) deceased donor kidney
transplant waitlist, and 20% of these patients had been waiting for at least six years [1]. These
stark figures reflect an ongoing shortage of donor kidneys and fuel interest in both expanding
the pool of potential donors and optimizing the use of available kidneys [2]. The majority of
deceased donor kidneys come from donors after brain death (DBD) who have died by neuro-
logic criteria. However, since their introduction in 1993, donors after circulatory death (DCD)
make up a growing proportion of all deceased donor kidneys. DCD kidneys comprised 2% of
deceased donor kidney transplants in 2000, 8% by 2005, and 20% by 2017 [3]. Use of DCD kid-
neys has also expanded among European transplant programs after the practice was approved
by the World Health Organization in 2011, but attitudes, policies, and practices vary geograph-
ically [4]. Of 35 European countries participating in a recent survey, 18 reported active DCD
programs and 9 additional countries reported interest in developing these programs [5].

In the US, the vast majority of DCD kidneys are obtained after a donor has died as defined
by absence of cardiopulmonary circulation after withdrawal of life-supporting treatment
(Maastricht category III) [6,7]. Transplant centers typically wait for a maximum of one hour
after withdrawal of life supporting treatment and are required to observe a two to five-minute
waiting period after cessation of cardiorespiratory function before death is declared [8]. Dur-
ing this waiting period, the donor can have systemic hypotension [8], which may cause ische-
mic kidney injury and likely contributes to the delayed graft function after transplant that
occurs for 50–60% of DCD recipients [9–11]. DCD kidneys have also been associated with lon-
ger hospital length of stay, readmissions, acute rejection, and more frequent graft loss com-
pared with DBD kidneys [10,12–14]. However, multiple recent studies suggest that despite a
higher risk of delayed graft function [15,16], DCD kidneys may offer comparable recipient
outcomes compared with DBD kidneys [1,11,17,18]. Nevertheless, DCD kidneys continue to
be considered lower quality than DBD kidneys in the US and Europe [5] and are discarded at
much higher rates than other deceased donor kidneys [1,19–21]. DCD status was determined
to be predictive of poorer graft survival as a component of the kidney donor risk index (KDRI)
by Rao et al. in a cohort of patients who received a kidney transplant from 1995 to 2005 and
the relevance of this factor was reaffirmed by Zhong et al. in a cohort of patients who received
a kidney transplant from 2000–2016 [22,23]. This index measure of kidney quality has become
an integral component of organ-matching and selection for the national kidney allocation
system.

The association between donor kidney DCD status and recipient outcomes remains
unclear. In this study, we compare rates of graft loss between DCD and DBD kidneys for a
contemporary cohort of deceased donor kidney transplant recipients.

## Patients and methods

### Study population and data source

We conducted a retrospective analysis of all adult kidney transplant recipients who underwent
their first single-organ deceased donor kidney transplant from 12/4/2014 to 6/30/2018 in the

United States. We used a dataset that was released by the Organ Procurement and Transplantation Network (OPTN) on 12/01/2018 and included data collected by the United Network for Organ Sharing (UNOS) through 9/30/2018. Recipients were excluded if they were younger than 18 years old, if they received simultaneous organ transplants, or if they received a repeat kidney transplant. For a secondary analysis, we created a comparator cohort of patients who received a deceased donor kidney transplant from 1/1/1995-12/3/2014 with the same exclusion criteria. The OPTN database is de-identified and publicly available; therefore, this study was exempt from human subject review as approved by the University of Washington Human Subjects Division.

Using UNOS donor data, we stratified cohort members into two groups based on whether they received a DCD or DBD kidney. We also collected data representing the annual number of DCD transplants performed in the US from 1995 to 2018, and the proportion of programs performing at least one DCD transplant per year.

## Covariates

Using data reported on UNOS transplant recipient forms, we determined donor age, sex, race or ethnicity (Asian, black, Hispanic, white, other), height, weight, cause of death (anoxia, cerebral vascular accident (CVA), head trauma, or other cause), serum creatinine, hepatitis C virus (HCV) positivity (by serology or NAT positivity) and history of diabetes mellitus (DM), hypertension, and cigarette smoking. We further determined recipient age, sex, race or ethnicity, height, weight, panel reactive antigen result (PRA), history of end-stage kidney disease (ESKD) prior to transplant, peripheral vascular disease, and malignancy, HCV and Epstein-Barr Virus (EBV) positivity, time spent on the waitlist, and primary cause of kidney disease (cancer, congenital kidney disease, diabetes mellitus, medication, glomerulonephritis, hypertension, metabolic disease, interstitial nephritis, polycystic kidney disease, systemic lupus erythematous, anatomic abnormality, or other cause). We also determined whether the donor graft was preserved by machine perfusion before transplantation, the graft cold ischemia time (CIT), and whether the graft was transplanted en-bloc (a technique in which two kidneys are transplanted together as one graft). We ascertained whether the donation occurred through organ sharing at the local, regional, or national level and the degree of HLA-DR and HLA-B antigen mismatching between donor and recipient. These factors were chosen to represent all currently available variables that were included in the foundational publication by Rao et al. that described characteristics associated with deceased donor graft function. [22] For a supplementary descriptive analysis, we ascertained the kidney donor profile index (KDPI) recorded for each deceased donor kidney.

There were 208 missing values for donor diabetes mellitus status, 258 missing values for donor hypertension status, 563 missing values for donor smoking status, and 7 missing values reporting whether the donor kidney was preserved by machine perfusion. All these missing values were recorded as unknown and entered into the analysis. 254 missing values for CIT were imputed by linear regression using distance, type of sharing, and region of transplant.

## Outcomes

We collected data for all recipients until 9/30/2018, which included 90 days beyond the end of study follow-up to account for late filing of UNOS transplant recipient forms. Our primary outcome of interest was all-cause graft loss (including graft loss secondary to recipient death) and secondary outcome was death-censored graft loss.

As a supplementary descriptive analysis, we used a modified version of the KDRI score that we calculated after excluding the coefficient for DCD status. [24] We then used KDRI-to-

KDPI mapping tables applicable to each year of transplant to approximate KDPI values. We compared this modified KDPI score with the KDPI reported by UNOS for DCD grafts received by cohort members.

## Statistical analysis

We used median and interquartile ranges (IQR) and Student's t-tests or Kruskal-Wallis tests, as appropriate for each distribution, for continuous variables and percentages and chi-square tests for categorical variables to describe and compare donor and recipient characteristics between groups defined by DCD versus DBD status (Table 1).

To determine the unadjusted and adjusted hazard ratio of graft loss associated with DCD status, we used univariable and multivariable Cox proportional hazards models with mixed effects. This model allowed us to account for cluster-specific random effects that result in differing baseline hazard functions between transplant centers [25]. For the multivariable analysis, we controlled for all measured donor characteristics (age, sex, race or ethnicity, height, weight, cause of death, serum creatinine, HCV status, history of DM, hypertension and smoking), recipient characteristics (age, sex, race or ethnicity, height, weight, PRA, history of ESKD, peripheral vascular disease, and malignancy, HCV and EBV positivity, time spent on the waitlist, and primary cause of kidney disease), and all measured aspects of transplant logistics (whether the graft was preserved by mechanical perfusion, CIT, whether the graft was transplanted en-bloc, sharing network [i.e., local, regional, national], and B- and DR-antigen mismatches).

In a supplementary analysis, we performed the same univariable and multivariable Cox proportional hazards analyses for an earlier cohort of patients who received a deceased donor transplant from 1/1/1995-12/3/2014. As in our primary analysis, we used transplant program as the random effects term and controlled for all measured donor and recipient characteristics and transplant logistics.

Results were considered statistically significant with a p-value less than 0.05. Prior to conducting this analysis, we estimated that with a sample size of 35,264 recipients, we would have an 80% power to detect a difference of 2.0% in the hazard ratio for graft loss associated with DCD compared with DBD status for the primary analysis. We performed comparative statistics using JMP-Pro Version 13.0.1 (SAS Institute, Inc., Cary, NC, USA) and Cox proportional hazards models using R version 3.5.1 and the coxme 2.2–10 package.

## Results

### Cohort characteristics

Our cohort consisted of 27,494 DBD kidney recipients and 7,770 DCD kidney recipients who received a kidney transplant in the US between 12/4/2014 and 6/30/2018. Recipients had a median follow up of 1.92 years (IQR 1.08–2.83). Compared with DBD donors, DCD donors were less frequently in the youngest and oldest age groups, and more often white (Table 1). The cause of death for DCD donors was more commonly anoxia or an unspecified other cause compared with DBD donors who more often died from a CVA or from head trauma. Serum creatinine was, on average, lower among DCD donors compared with DBD donors, and DCD donors less often had a positive HCV status. DCD donors more often had a history of smoking, but less often had a history of diabetes or hypertension compared with DBD donors. DCD recipients were more often white and more frequently had a low PRA. DCD grafts were more often preserved by machine perfusion before transplantation, had a longer CIT, and more often came from local sharing networks compared with DBD grafts.

**Table 1. Cohort characteristics.**

| | Deceased donor graft type | | |
| --- | --- | --- | --- |
| | Donor after brain death (n = 27,494) | Donor after circulatory death (n = 7,770) | p-value* |
| **Donor characteristics** | | | |
| Age groups, % | | | |
| 0–17 y | 9.1 | 7.8 | <0.001 |
| 18–30 y | 25.1 | 23.6 | 0.005 |
| 31–45 y | 28.9 | 30.0 | 0.06 |
| 46–65 y | 34.1 | 38.6 | <0.001 |
| >65 y | 2.8 | 0.1 | <0.001 |
| Female, % | 40.5 | 34.3 | <0.001 |
| Race, % | | | |
| Asian | 2.6 | 1.9 | <0.001 |
| Black | 15.4 | 8.0 | <0.001 |
| Hispanic | 14.9 | 9.8 | <0.001 |
| White | 64.8 | 79.1 | <0.001 |
| Other | 2.3 | 1.2 | <0.001 |
| Height groups, % | | | |
| <80 cm | 1.0 | 1.0 | 0.61 |
| 80 to 170 cm | 47.6 | 40.8 | <0.001 |
| >170 cm | 51.4 | 58.3 | <0.001 |
| Weight groups, % | | | |
| <30 kg | 4.2 | 2.7 | <0.001 |
| 30 to 80 kg | 47.5 | 43.9 | <0.001 |
| >80 to 110 kg | 37.8 | 40.5 | <0.001 |
| >110 kg | 10.5 | 13.0 | <0.001 |
| Cause of death, % | | | |
| Anoxia | 38.1 | 51.4 | <0.001 |
| CVA | 27.9 | 16.4 | <0.001 |
| Head Trauma | 31.2 | 27.4 | <0.001 |
| Other | 2.8 | 4.9 | <0.001 |
| Serum Creatinine (mg/dL), median (IQR) | 1.0 (0.7–1.5) | 0.8 (0.6–1.1) | <0.001 |
| HCV status positive*, % | 4.8 | 2.2 | <0.001 |
| Diabetes, % | 7.7 | 6.0 | <0.001 |
| Hypertension, % | 29.0 | 26.7 | <0.001 |
| Cigarette Smoking, % | 18.7 | 19.8 | 0.03 |
| **Recipient characteristics** | | | |
| Age, years (IQ) | 55 (43–63) | 56 (46–64) | <0.001 |
| Female, % | 40.9 | 39.2 | 0.006 |
| Race, % | | | |
| Asian | 7.4 | 7.5 | 0.83 |
| Black | 37.1 | 32.4 | <0.001 |
| Hispanic | 19.8 | 18.9 | 0.07 |
| White | 33.2 | 38.1 | <0.001 |
| Other | 2.4 | 3.1 | <0.001 |
| Height groups, % | | | |
| <163 cm | 24.4 | 24.1 | 0.50 |
| 163–178 cm | 55.3 | 54.4 | 0.17 |
| >178 | 20.3 | 21.5 | 0.02 |

(*Continued*)

**Table 1.** (Continued)

| | Deceased donor graft type | | |
|---|---|---|---|
| | Donor after brain death (n = 27,494) | Donor after circulatory death (n = 7,770) | p-value* |
| **Donor characteristics** | | | |
| Weight groups, % | | | |
| <70 kg | 28.3 | 26.0 | <0.001 |
| 70–85 kg | 31.4 | 31.3 | 0.88 |
| >85–100 kg | 23.2 | 24.6 | 0.01 |
| >100 kg | 17.1 | 18.1 | 0.04 |
| Panel reactive antibody level | | | |
| 0–10 | 68.8 | 71.0 | <0.001 |
| 11–100 | 31.2 | 29.0 | <0.001 |
| End stage kidney disease | 88.3 | 87.8 | 0.16 |
| Peripheral vascular disease | 8.8 | 9.2 | 0.36 |
| History of malignancy | 8.1 | 7.7 | 0.25 |
| HCV Positive | 6.1 | 4.3 | <0.001 |
| EBV Positive | 87.4 | 87.5 | 0.77 |
| Time on waiting on list, days (IQR) | 698 (194–1433) | 692 (191–1416) | 0.20 |
| Primary cause of kidney disease | | | |
| Cancer | 0.5 | 0.6 | 0.63 |
| Congenital | 1.2 | 1.2 | 0.87 |
| Diabetes Mellitus | 30.8 | 33.4 | <0.001 |
| Medication | 0.3 | 0.2 | 0.05 |
| Glomerulonephritis | 10.6 | 10.6 | 0.86 |
| Hypertension | 28.3 | 25.8 | <0.001 |
| Metabolic disease | 1.2 | 1.3 | 0.42 |
| Interstitial nephritis | 6.3 | 6.9 | 0.05 |
| Polycystic kidney disease | 7.9 | 8.1 | 0.59 |
| Systemic lupus erythematous | 3.5 | 2.9 | 0.01 |
| Anatomic abnormality | 1.2 | 1.1 | 0.47 |
| Other | 8.2 | 8.1 | 0.86 |
| **Transplant logistics** | | | |
| Kidney graft placed on machine perfusion | 29.8 | 51.8 | <0.001 |
| Cold ischemia time, hrs, median (IQR) | 16.2 (11.0–22.3) | 18.7 (14.0–23.5) | <0.001 |
| En-bloc kidney graft | 1.9 | 1.6 | 0.03 |
| Sharing network, % | | | |
| Local | 72.0 | 76.5 | <0.001 |
| Regional | 13.2 | 11.2 | <0.001 |
| National | 14.8 | 12.3 | <0.001 |
| Number of B-antigen mismatches, % | | | |
| 0 | 6.4 | 6.2 | 0.54 |
| 1 | 24.5 | 25.4 | 0.09 |
| 2 | 69.2 | 68.4 | 0.21 |
| Number of DR-antigen mismatches, % | | | |
| 0 | 14.7 | 15.5 | 0.09 |
| 1 | 48.4 | 49.3 | 0.17 |
| 2 | 36.8 | 35.2 | 0.01 |

* by serology or NAT positivity.

P-value by Student's t-tests, Kruskal-Wallis tests, or chi-squared as appropriate.

## Outcomes

Overall, the hazard ratio for graft loss associated with DCD status compared with DBD status was 0.96 (95% confidence interval (CI) 0.87–1.05) in univariable analysis and 1.03 (95% CI 0.94–1.13) in multivariable analysis (Table 2). The hazard ratio for death-censored graft loss associated with DCD status compared with DBD status was 0.97 (95% CI 0.91–1.06) in univariable analysis and 1.05 (95% CI 0.99–1.11) in multivariable analysis. In sensitivity analyses, we found no significant interaction effects for any variables included in the model and no difference in results after excluding imputed CIT values.

For the earlier cohort of patients who received a deceased donor kidney transplant from 1995–2014, the hazard ratio for graft loss associated with DCD status compared with DBD status was 0.93 (95% CI 0.91–0.96) in univariable analysis and 1.01 (95% CI 0.98–1.04) in multivariable analysis. The hazard ratio for death-censored graft loss was 0.99 (0.96–1.03) in univariable and 1.11 (1.07–1.15) in multivariable analysis.

Among 7,770 DCD donor kidneys transplanted from 2014–2018, 961 (12.4%) had a KDPI of 0–20%, 6248 (80.4%) had a KDPI of 21–85%, and 561 (7.2%) had a KDPI of 86–100% (Table 3). After modifying the published KDRI model by excluding the coefficient for DCD status [24], we calculated that 2,219 (28.6%) members of our cohort would have had a KDPI of 0–20%, 5,445 (70.1%) would have had a KDPI of 21–85%, and 106 (1.4%) would have had a KDPI of 86–100% if the DCD status coefficient were not included in the KDRI model.

## Discussion

For a national cohort of adults in the US who received a deceased donor kidney transplant between 2014 and 2018, we did not find a difference in the likelihood of experiencing all-cause or death-censored graft loss for DCD kidney recipients compared with DBD kidney recipients, even after adjustment for measured differences in donor and recipient characteristics and transplant logistics.

**Table 2. Association of DCD donor status with kidney graft loss.**

|  | HR (95% CI) | p-value |
|---|---|---|
| **Contemporary cohort: 2014–2018** |  |  |
| Graft loss |  |  |
| Univariable analysis | 0.96 (0.87–1.05) | 0.35 |
| Multivariable analysis | 1.03 (0.95–1.13) | 0.57 |
| Death-censored graft loss |  |  |
| Univariable analysis | 0.97 (0.91–1.06) | 0.66 |
| Multivariable analysis | 1.05 (0.99–1.11) | 0.42 |
| **Early cohort: 1995–2014** |  |  |
| Graft loss |  |  |
| Univariable analysis | 0.93 (0.91–0.96) | 0.54 |
| Multivariable analysis | 1.01 (0.98–1.04) | 0.77 |
| Death-censored graft loss |  |  |
| Univariable analysis | 0.99 (0.96–1.03) | 0.88 |
| Multivariable analysis | 1.11 (1.07–1.15) | 0.003 |

Cox proportional hazard model with mixed effects using 'transplant program' as the random variable. Multivariable analyses are controlled for all measured donor variables, all measured recipient characteristics, and transplant logistic.

Abbreviations: DCD; Donor after circulatory death; HR, hazard Ratio; CI, confidence interval.

**Table 3. KDPI of DCD kidneys transplanted from 2014–2018 using KDPI models with and without a DCD term.**

| KDPI group, n (%) | KDPI model including DCD coefficient (current model) | KDPI model excluding DCD coefficient |
|---|---|---|
| 0–20% | 961 (12.4) | 2,219 (28.6) |
| 21–85% | 6,248 (80.4) | 5,445 (70.1) |
| 86–100% | 561 (7.2) | 106 (1.4) |

Abbreviations: DCD, donor after circulatory death; KDPI, kidney donor profile index.

In the context of ongoing organ shortage, the kidney transplant community has striven to innovate and maximize use of available donor grafts while also respecting an obligation to preserve excellent outcomes for individual patients. Recognizing the tension between these two goals, there has been caution in accepting DCD kidneys in light of early studies showing relatively poor outcomes compared with DBD kidneys [10,12–14,22,23]. However, for a contemporary cohort of deceased donor kidney transplant recipients, we did not find a difference in risk of graft loss between DCD and DBD kidney recipients. These findings call into question the assumption that these two types of donor kidneys differ in quality. Two major distinctions between our analysis and existing reports may contribute to these differing results. First, we used a novel approach to modeling the likelihood of graft loss that accounted for possible clustering of unmeasured differences in baseline hazard functions between transplant programs by using a Cox proportional hazards model with mixed effects [25]. Differences in experience, technique, and practice patterns between transplant centers may meaningfully contribute to estimates of outcomes for DCD versus DBD grafts. This multilevel approach to survival analysis may prove valuable for future analyses modeling other kidney transplant outcomes. Second, we studied a more contemporary cohort of transplant recipients compared with prior reports. Our supplementary analysis offered mixed signals about whether the risk of graft loss associated with DCD compared with DBD status might differ between early and more recent time periods. Specifically, our multivariable analysis for an earlier cohort of transplant recipients did not show a difference between DCD and DBD recipients for the hazard of all-cause graft loss, but we did detect a greater hazard for death-censored graft loss associated with DCD status for this earlier cohort. This signal of improving DCD graft survival compared with DBD graft survival may reflect the great strides that the transplant community has made over the last decades in effective use of DCD kidneys including better selection of potential donors [16,18,26–28], standardized surgical technique [8], and improved preservation of kidney grafts [29,30].

The inclusion of DCD status as a marker of graft quality in the KDPI may have a substantial impact on distribution and discard patterns of deceased donor kidneys in the US. The current US kidney allocation system relies on the KDPI to support a longevity-matching strategy whereby the highest-quality grafts (KDPI less than 20%) are allocated to patients with the longest post-transplant estimated survival [31]. At the other end of the kidney quality spectrum, discard rates for kidneys with a KDPI greater than 85% have substantially increased since implementation of the new kidney allocation system [1]. This might be attributed in part to what Bae et al. describe as a "labeling effect" that can bias transplant teams toward discard of high-KDPI kidneys [32]. Our supplementary analysis suggests that inclusion of DCD status in the KDPI may materially impact how these donor kidneys are categorized. For example, a kidney from a 65-year-old donor with no pertinent medical history who died from head trauma

would be assigned a KDPI of 85% if the donor died by DCD criteria, and 74% if the donor died by DBD criteria [33]. Similarly, a kidney from a 25-year-old donor with no pertinent medical history who died from head trauma would be assigned a KDPI of 31% if the donor died by DCD criteria and 17% if the donor died by DBD criteria. After excluding the DCD term from the KDRI model, 1,258 (16%) DCD kidneys in our cohort were re-classified from a KDPI of 20%-85% to ≤20%, and 455 (6%) DCD kidneys in our cohort were re-classified from a KDPI of >85% to ≤85%. While we intend this analysis only as an illustrative approximation, it suggests that there may be a sizable number of high-quality DCD kidneys more effectively allocated to recipients who have a longer post-transplant estimated survival and that there may be many good-quality DCD kidneys at risk of being discarded. These observations underline the importance of re-evaluating the inclusion of DCD status in the KDPI and signal an opportunity for transplant center teams to identify and utilize potentially valuable DCD kidneys based on their assessment of multiple graft characteristics rather than relying primarily on an aggregate measure of quality.

## Limitations

This study should be interpreted with several limitations in mind. First, because we targeted a contemporary cohort of transplant recipients, the follow-up period is relatively short. Future study may offer insight into patterns in longer-term graft survival and also lend additional power to detect small differences in graft survival. Second, kidney graft quality may be judged by other patient-important outcomes that we are unable to report using this study design (e.g., acute rejection, length of hospital stay, rehospitalizations). However, our findings do align with accumulating observations from other work showing little difference in other outcomes for recipients of DCD compared with DBD grafts in the US and internationally [15–18,34]. Third, OPTN registry data is limited to outcomes for grafts that were selected for transplant, so these findings may not apply to discarded kidneys. Although we adjusted for all measured characteristics, there may be unmeasured differences between groups. Finally, estimation of the KDPI distribution for our cohort using a KDRI model without the DCD coefficient is intended only as an illustrative approximation. Definitive exclusion of the DCD coefficient might involve adjusting the weight of other coefficients included in the KDRI model.

## Conclusions

For a contemporary cohort of deceased donor kidney transplant recipients in the US, we did not observe a significant difference in risk of graft loss for DCD compared with DBD kidney recipients. These findings signal uncertainty about whether DCD status independently contributes to measures of graft quality, and identify an opportunity to promote more effective use of scarce deceased donor kidneys by continuing to update our understanding of markers of graft quality.

## Acknowledgments

The interpretation and reporting of these data are the responsibility of the authors and in no way should be seen as an official policy of or interpretation by the OPTN or the U.S. Government.

## Author Contributions

**Conceptualization:** Catherine R. Butler, James D. Perkins, Christopher K. Johnson, Christopher D. Blosser, Iris De Castro, Nicolae Leca, Lena Sibulesky.

**Data curation:** James D. Perkins.

**Formal analysis:** Catherine R. Butler, James D. Perkins, Lena Sibulesky.

**Methodology:** James D. Perkins.

**Project administration:** Catherine R. Butler.

**Supervision:** Lena Sibulesky.

**Validation:** James D. Perkins.

**Visualization:** Catherine R. Butler.

**Writing – original draft:** Catherine R. Butler, James D. Perkins, Christopher D. Blosser, Lena Sibulesky.

**Writing – review & editing:** Catherine R. Butler, James D. Perkins, Christopher K. Johnson, Iris De Castro, Nicolae Leca, Lena Sibulesky.

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
