## [Decision Letter · Decision Letter 0]

13 Nov 2019

PONE-D-19-29374

Contemporary patterns in kidney graft survival from donors after circulatory death in the United States

PLOS ONE

Dear Dr. Sibulesky,

Thank you for submitting your manuscript to PLOS ONE. After careful consideration, we feel that it has merit but does not fully meet PLOS ONE’s publication criteria as it currently stands. Therefore, we invite you to submit a revised version of the manuscript that addresses the points raised during the review process.

ACADEMIC EDITOR: 

This MS is of interest to the tx community, showing equivalent survival of DCD vs DBD kidney transplantation.

Reviewers do highlight some issues that would need to be addressed in detail. Especially, some additional outcomes are missing that would be important to include.

We would appreciate receiving your revised manuscript by Dec 28 2019 11:59PM. To enhance the reproducibility of your results, we recommend that if applicable you deposit your laboratory protocols in protocols.io, where a protocol can be assigned its own identifier (DOI) such that it can be cited independently in the future. For instructions see: http://journals.plos.org/plosone/s/submission-guidelines#loc-laboratory-protocols

We look forward to receiving your revised manuscript.

Kind regards,

Frank JMF Dor, M.D., Ph.D., FEBS, FRCS

Academic Editor

PLOS ONE

Journal Requirements:

The interpretation and reporting of these data are the responsibility of the authors and in no way should be seen as an official policy of or interpretation by the OPTN or the U.S. Government. Dr. Butler is supported by a training grant from the NIDDK (5T32DK007467-33).

4. To comply with PLOS ONE submission guidelines, in your Methods section, please provide additional information regarding your statistical analyses. For more information on PLOS ONE's expectations for statistical reporting, please see https://journals.plos.org/plosone/s/submission-guidelines.#loc-statistical-reporting.

Reviewers' comments:

Reviewer's Responses to Questions

**Comments to the Author**

1. Is the manuscript technically sound, and do the data support the conclusions?

Reviewer #1: Yes

Reviewer #2: Yes

2. Has the statistical analysis been performed appropriately and rigorously? 

Reviewer #1: Yes

Reviewer #2: No

3. Have the authors made all data underlying the findings in their manuscript fully available?

Reviewer #1: Yes

Reviewer #2: Yes

4. Is the manuscript presented in an intelligible fashion and written in standard English?

Reviewer #1: Yes

Reviewer #2: Yes

5. Review Comments to the Author

Reviewer #1: Contemporary patterns in kidney graft survival from donors after circulatory death in the United States

Very interesting and important study.

Limitations were stated, including the short term follow up and the need for future studies with longer follow up.

1. Could I ask you please if you have any data on the type of DCD donor eg controlled or uncontrolled or Maastricht category? If yes, did you do

any analysis according to these? If not, please state in the paper that no such data were available.

2. Do you have any data on delayed graft function to add to the study?

3. Please add the current practice followed in the US with respect to DCD kidney donation:

a. How many minutes is the “waiting period” (time after cessation of cardiorespiratory function before death is declared)?

b. How long do you wait before standing down if there is no cessation of cardiorespiratory function? Does this practice very between US centers?

Thank you.

Reviewer #2: This paper describes the association of DCD versus DBD donor status with all cause graft loss after kidney transplantation in a recent US-cohort of transplants performed between April 2014 and September 2017. The authors found no association of DCD status with graft loss.

As many transplant professionals perceive DCD kidneys as inferior and the KDRI is negatively impacted by the DCD donor status this paper has a potentially important message for the acceptance of offered kidneys and allocation policy making. The paper confirms recent papers in outcomes in other countries showing similar outcomes in DCD transplantation as compared to DBD transplantation. Overall the central message is clear and relevant. I do feel that somewhat more effort could have been made to corroborate the findings.

Some comments:

- The median follow up in the study is below one year, thus effects of donor status on longer term outcome may be missed. Follow up is short due to the study design in which the authors aim to analyse a recent cohort. As it has been suggested that DCD donor status negatively impacts early outcome and this is compensated by improved long term outcome, the short term outcome described in this study is probably highly relevant. However, as the authors explain their findings by improved management in DCD transplantation, it may be helpful to actually demonstrate that this short term outcome has improved compared to earlier cohorts.

- Unfortunately data on other short term outcomes such as delayed graft function or primary non function are not given. Insight into these parameters would be helpful to explain the findings of the study. Again a comparison with earlier cohorts would make the findings more robust.

- While all cause graft loss as given in the study is the outcome of primary interest, a separate analysis of death censored graft survival may give some additional insight into changing patterns over time.

Minor points:

- A recent study on the Dutch DCD experience confirming the equivalence of graft survival in DCD and DBD kidney transplantation would seem a useful reference. (Schaapherder et al. EClinicalMedicine. 2018 Oct 9;4-5:25-31)

- The following is stated in the introduction: “A requisite two to five-minute “waiting period” after cessation of cardiorespiratory function and before death is declared can cause systemic hypotension..” “Can cause systemic hypotension” is a somewhat awkward description of the risks after after cessation of circulation.

6. PLOS authors have the option to publish the peer review history of their article (what does this mean?). If published, this will include your full peer review and any attached files.

Reviewer #1: Yes: Nicos Kessaris

Reviewer #2: No

---

## [Author Response · Author response to Decision Letter 0]

4 Mar 2020

Editorial Staff: 

i) We note that you have indicated that data from this study are available upon request. PLOS only allows data to be available upon request if there are legal or ethical restrictions on sharing data publicly. For information on unacceptable data access restrictions, please see http://journals.plos.org/plosone/s/data-availability#loc-unacceptable-data-access-restrictions.

If there are ethical or legal restrictions on sharing a de-identified data set, please explain them in detail (e.g., data contain potentially identifying or sensitive patient information) and who has imposed them (e.g., an ethics committee). Please also provide contact information for a data access committee, ethics committee, or other institutional body to which data requests may be sent.

Data used in this study is managed by the Scientific Registry of Transplant Recipients. It is publicly available upon request at: https://www.srtr.org/requesting-srtr-data/data-requests/

ii) Thank you for stating the following in the Acknowledgments Section of your manuscript:

The interpretation and reporting of these data are the responsibility of the authors and in no way should be seen as an official policy of or interpretation by the OPTN or the U.S. Government. Dr. Butler is supported by a training grant from the NIDDK (5T32DK007467-33).

We have removed any reference to funding from the manuscript. The updated funding statement should read: 

Dr. Butler is supported by a training grant from the NIDDK (5T32DK007467-33).

2. Please remove your figures/ from within your manuscript file, leaving only the individual TIFF/EPS image files. These will be automatically included in the reviewer’s PDF

This has been corrected. 

Academic Editor:

This MS is of interest to the tx community, showing equivalent survival of DCD vs DBD kidney transplantation.

Reviewers do highlight some issues that would need to be addressed in detail. Especially, some additional outcomes are missing that would be important to include.

Below, we include a point-by-point response to reviewer comments including regarding alternative outcomes. 

Reviewer #1:

1. Could I ask you please if you have any data on the type of DCD donor eg controlled or uncontrolled or Maastricht category? If yes, did you do any analysis according to these? If not, please state in the paper that no such data were available.

We have now clarified that all donations in the US are controlled (Maastricht category III). [Introduction, paragraph 2]

2. Do you have any data on delayed graft function to add to the study?

We agree that delayed graft function would be a valuable outcome. However, because definitions of delayed graft function vary between transplant centers and reporting is heterogenous, we did not believe that the current dataset offered sufficiently robust information to include this outcome for the current analysis. 

3. Please add the current practice followed in the US with respect to DCD kidney donation:

a. How many minutes is the “waiting period” (time after cessation of cardiorespiratory function before death is declared)?

We now describe the 2-5 minute “waiting period” before declaration of death for DCD donors. [Introduction, paragraph 2]

b. How long do you wait before standing down if there is no cessation of cardiorespiratory function? Does this practice very between US centers?

We now include information about what is typically a 1-hour cut-off time after withdrawal of life-supporting treatment in the US. [Introduction, paragraph 2]

Reviewer #2: 

This paper describes the association of DCD versus DBD donor status with all cause graft loss after kidney transplantation in a recent US-cohort of transplants performed between April 2014 and September 2017. The authors found no association of DCD status with graft loss.

As many transplant professionals perceive DCD kidneys as inferior and the KDRI is negatively impacted by the DCD donor status this paper has a potentially important message for the acceptance of offered kidneys and allocation policy making. The paper confirms recent papers in outcomes in other countries showing similar outcomes in DCD transplantation as compared to DBD transplantation. Overall the central message is clear and relevant. I do feel that somewhat more effort could have been made to corroborate the findings.

Some comments:

- The median follow up in the study is below one year, thus effects of donor status on longer term outcome may be missed. Follow up is short due to the study design in which the authors aim to analyse a recent cohort. As it has been suggested that DCD donor status negatively impacts early outcome and this is compensated by improved long term outcome, the short term outcome described in this study is probably highly relevant. However, as the authors explain their findings by improved management in DCD transplantation, it may be helpful to actually demonstrate that this short term outcome has improved compared to earlier cohorts.

We have now extended the follow-up period for our cohort as updated data was available in the time since initial submission. We also include a secondary analysis for an earlier cohort in order to support more direct comparisons across eras. We chose to define this earlier cohort with a longer follow-up time to assess whether our choice of model (a mixed cox proportional hazard model) may also contribute to differences that we see between our analyses and prior studies. 

- Unfortunately data on other short term outcomes such as delayed graft function or primary non function are not given. Insight into these parameters would be helpful to explain the findings of the study. Again a comparison with earlier cohorts would make the findings more robust.

As noted in response to Reviewer #1’s comments above, we agree that delayed graft function is an outcome of general interest. We decided not to pursue this outcome because we did not find reports in our dataset to be sufficiently consistent to support validity. 

- While all cause graft loss as given in the study is the outcome of primary interest, a separate analysis of death censored graft survival may give some additional insight into changing patterns over time.

We decided to not include death-censored graft loss for this analysis because we find it difficult to reliably disentangle what is often a complex relationship between death and deteriorating graft function. For example, failing graft function may nonetheless lead to recipient death before frank graft failure can be reported. We believe that this approach aligns with existing work such as in Rao et al. and Zhong et al.

Minor points:

- A recent study on the Dutch DCD experience confirming the equivalence of graft survival in DCD and DBD kidney transplantation would seem a useful reference. (Schaapherder et al. EClinicalMedicine. 2018 Oct 9;4-5:25-31)

We appreciate the suggestion and now reference this study. [Discussion, paragraph 2]

- The following is stated in the introduction: “A requisite two to five-minute “waiting period” after cessation of cardiorespiratory function and before death is declared can cause systemic hypotension..” “Can cause systemic hypotension” is a somewhat awkward description of the risks after after cessation of circulation.

We have clarified this language. [Introduction, paragraph 2]

---

## [Decision Letter · Decision Letter 1]

2 Apr 2020

PONE-D-19-29374R1

Contemporary patterns in kidney graft survival from donors after circulatory death in the United States

PLOS ONE

Dear Dr. Sibulesky,

Thank you for submitting your manuscript to PLOS ONE. After careful consideration, we feel that it has merit but does not fully meet PLOS ONE’s publication criteria as it currently stands. Therefore, we invite you to submit a revised version of the manuscript that addresses the points raised during the review process.

ACADEMIC EDITOR: 

Thank you for making an effort to address the issues pointed out by the reviewers. However, I agree with reviewer 2 that a few remaining issues should be dealt with appropriately. I hope you can take those points seriously and follow them through in the discussion and conclusions of a revised MS. I would like to emphasize that the points brought up by reviewer 2 are crucial and the revised MS will only be taken into consideration for publication if the revisions are satisfactory. I think your study can really contribute to the field if these extremely important concerns can be ironed out. Looking forward to your revised MS.

We would appreciate receiving your revised manuscript by May 17 2020 11:59PM. To enhance the reproducibility of your results, we recommend that if applicable you deposit your laboratory protocols in protocols.io, where a protocol can be assigned its own identifier (DOI) such that it can be cited independently in the future. For instructions see: http://journals.plos.org/plosone/s/submission-guidelines#loc-laboratory-protocols

We look forward to receiving your revised manuscript.

Kind regards,

Frank JMF Dor, M.D., Ph.D., FEBS, FRCS

Academic Editor

PLOS ONE

Reviewers' comments:

Reviewer's Responses to Questions

**Comments to the Author**

1. If the authors have adequately addressed your comments raised in a previous round of review and you feel that this manuscript is now acceptable for publication, you may indicate that here to bypass the “Comments to the Author” section, enter your conflict of interest statement in the “Confidential to Editor” section, and submit your "Accept" recommendation.

Reviewer #1: All comments have been addressed

Reviewer #2: (No Response)

2. Is the manuscript technically sound, and do the data support the conclusions?

Reviewer #1: Yes

Reviewer #2: No

3. Has the statistical analysis been performed appropriately and rigorously? 

Reviewer #1: Yes

Reviewer #2: Yes

4. Have the authors made all data underlying the findings in their manuscript fully available?

Reviewer #1: Yes

Reviewer #2: Yes

5. Is the manuscript presented in an intelligible fashion and written in standard English?

Reviewer #1: Yes

Reviewer #2: Yes

6. Review Comments to the Author

Reviewer #1: Thank you for addressing all the questions and comments made previously and for providing the extra US data for the period between 1995 and 2014. It would be interesting to repeat the study in a few years time and compare the outcomes of DBD and DCD grafts at 3 and 5 years post transplant.

Reviewer #2: I do not feel that the reviewers comments have been adequately addressed:

- The authors now show that with their methodology the multivariate analysis does not show a survival disadvantage in the historical cohort. This is a direct contradiction with the central message of the paper and has not lead to any adaptations in the discussion or the conclusions that still ponder on improved outcome. Either DCD transplantation outcome was not poorer in the past or the applied analysis is not able to detect this.

- The authors dismiss the suggestion to include death censored graft loss by citing two papers on the development of a risk index and making the point that death censored graft loss misses patient death due to poor graft function as an outcome. While death censored graft loss of course had its shortcomings the same holds true for overall composite graft loss. I do not think that papers aiming to develop a risk index for patient survival after transplantation are adequate references to justify leaving out death censored graft survival in a paper on understanding changing outcome patterns in DCD transplantation.

7. PLOS authors have the option to publish the peer review history of their article (what does this mean?). If published, this will include your full peer review and any attached files.

Reviewer #1: Yes: Nicos Kessaris

Reviewer #2: No

---

## [Author Response · Author response to Decision Letter 1]

13 Apr 2020

Response to editorial and reviewer comments:

Academic editor:

Thank you for making an effort to address the issues pointed out by the reviewers. However, I agree with reviewer 2 that a few remaining issues should be dealt with appropriately. I hope you can take those points seriously and follow them through in the discussion and conclusions of a revised MS. I would like to emphasize that the points brought up by reviewer 2 are crucial and the revised MS will only be taken into consideration for publication if the revisions are satisfactory. I think your study can really contribute to the field if these extremely important concerns can be ironed out. Looking forward to your revised MS.

As detailed below, we added death-censored graft loss as a secondary outcome and made major changes to the discussion to reflect the reviewer’ comments. 

Reviewer #1: Thank you for addressing all the questions and comments made previously and for providing the extra US data for the period between 1995 and 2014. It would be interesting to repeat the study in a few years time and compare the outcomes of DBD and DCD grafts at 3 and 5 years post transplant.

We agree with suggestions for future studies and thank the reviewer for their review. 

Reviewer #2: 

- The authors now show that with their methodology the multivariate analysis does not show a survival disadvantage in the historical cohort. This is a direct contradiction with the central message of the paper and has not lead to any adaptations in the discussion or the conclusions that still ponder on improved outcome. Either DCD transplantation outcome was not poorer in the past or the applied analysis is not able to detect this.

We fully appreciate the reviewer’s critique and have made major changes to re-focus the discussion around the novel methodology (Cox proportional hazard model with mixed effects using transplant center as the random effect term) that we believe contributes to differences in our findings compared with earlier reports. We have also removed a figure quantifying the changing volume of DCD kidney transplants in the US to better focus discussion on this first point of distinction between our study and earlier work. However, in performing the suggested analyses for death-censored graft loss on the earlier cohort, we now report higher risk of death-censored graft loss for DCD kidneys in this earlier cohort. In light of this new finding, we retain a much attenuated discussion of possible improvements in practice around DCD transplant that could have contributed to different findings for early and contemporary cohorts of deceased donor transplant recipients. 

- The authors dismiss the suggestion to include death censored graft loss by citing two papers on the development of a risk index and making the point that death censored graft loss misses patient death due to poor graft function as an outcome. While death censored graft loss of course had its shortcomings the same holds true for overall composite graft loss. I do not think that papers aiming to develop a risk index for patient survival after transplantation are adequate references to justify leaving out death censored graft survival in a paper on understanding changing outcome patterns in DCD transplantation.

We take the reviewers point, and have added death-censored graft loss as a secondary outcome. For the primary (contemporary) cohort of transplant recipients, findings are similar for the outcomes of all-cause and death-censored graft loss.

---

## [Decision Letter · Decision Letter 2]

29 Apr 2020

PONE-D-19-29374R2

Contemporary patterns in kidney graft survival from donors after circulatory death in the United States

PLOS ONE

Dear Dr. Sibulesky,

Thank you for submitting your manuscript to PLOS ONE. After careful consideration, we feel that a minor revision would still be required. Therefore, we invite you to submit a revised version of the manuscript that addresses the points raised during the review process.

Thank you for making the changes to your paper as requested. The MS has significantly improved thanks to this. I would like to propose to add reviewer 1's suggestion to the discussion. Provisionally accepted pending the minor revision.

We would appreciate receiving your revised manuscript by Jun 13 2020 11:59PM. To enhance the reproducibility of your results, we recommend that if applicable you deposit your laboratory protocols in protocols.io, where a protocol can be assigned its own identifier (DOI) such that it can be cited independently in the future. For instructions see: http://journals.plos.org/plosone/s/submission-guidelines#loc-laboratory-protocols

We look forward to receiving your revised manuscript.

Kind regards,

Frank JMF Dor, M.D., Ph.D., FEBS, FRCS

Academic Editor

PLOS ONE

Reviewers' comments:

Reviewer's Responses to Questions

**Comments to the Author**

1. If the authors have adequately addressed your comments raised in a previous round of review and you feel that this manuscript is now acceptable for publication, you may indicate that here to bypass the “Comments to the Author” section, enter your conflict of interest statement in the “Confidential to Editor” section, and submit your "Accept" recommendation.

Reviewer #1: (No Response)

Reviewer #2: All comments have been addressed

2. Is the manuscript technically sound, and do the data support the conclusions?

Reviewer #1: Yes

Reviewer #2: Yes

3. Has the statistical analysis been performed appropriately and rigorously? 

Reviewer #1: Yes

Reviewer #2: Yes

4. Have the authors made all data underlying the findings in their manuscript fully available?

Reviewer #1: Yes

Reviewer #2: Yes

5. Is the manuscript presented in an intelligible fashion and written in standard English?

Reviewer #1: Yes

Reviewer #2: Yes

6. Review Comments to the Author

Reviewer #1: Thank you for making the changes suggested by the second reviewer. The large sample of DBD and DCD donors as well as the interesting outcomes, contribute hugely to the value of this paper.

After reading your publication, members of the transplant community will be questioning whether they should be accepting more DCD kidneys. Can I ask you to add your thoughts on this issue and how you may have changed your practice following these outcomes please?

Reviewer #2: I want to thank the authors for the latest changes to the paper. The current version has a clear and important message. I have no further comments.

7. PLOS authors have the option to publish the peer review history of their article (what does this mean?). If published, this will include your full peer review and any attached files.

Reviewer #1: Yes: Nicos Kessaris

Reviewer #2: Yes: Stefan P. Berger

---

## [Author Response · Author response to Decision Letter 2]

7 May 2020

Response to editorial and reviewer comments:

Academic editor:

Thank you for making the changes to your paper as requested. The MS has significantly improved thanks to this. I would like to propose to add reviewer 1's suggestion to the discussion. Provisionally accepted pending the minor revision

Thank you, we have modified to include Reviewer 1’s comments as below.

Reviewer #1: Thank you for making the changes suggested by the second reviewer. The large sample of DBD and DCD donors as well as the interesting outcomes, contribute hugely to the value of this paper.

After reading your publication, members of the transplant community will be questioning whether they should be accepting more DCD kidneys. Can I ask you to add your thoughts on this issue and how you may have changed your practice following these outcomes please?

As clinicians, we do believe that this study and others encourage us to consider accepting DCD kidney grafts more readily if KDPI is high but other graft characteristics are favorable. We have added this consideration to the discussion.

Reviewer #2: I want to thank the authors for the latest changes to the paper. The current version has a clear and important message. I have no further comments.

We greatly appreciate the reviewer’s comments, which we believe have enhanced the manuscript.

---

## [Editor Report · Decision Letter 3]

11 May 2020

Contemporary patterns in kidney graft survival from donors after circulatory death in the United States

PONE-D-19-29374R3

Dear Dr. Sibulesky,

We are pleased to inform you that your manuscript has been judged scientifically suitable for publication and will be formally accepted for publication once it complies with all outstanding technical requirements.

With kind regards,

Frank JMF Dor, M.D., Ph.D., FEBS, FRCS

Academic Editor

PLOS ONE
---

## [Editor Report · Acceptance letter]

15 May 2020

PONE-D-19-29374R3 

Contemporary patterns in kidney graft survival from donors after circulatory death in the United States 

Dear Dr. Sibulesky:

I am pleased to inform you that your manuscript has been deemed suitable for publication in PLOS ONE. Congratulations! Your manuscript is now with our production department. 

With kind regards,

on behalf of

Dr. Frank JMF Dor 

Academic Editor

PLOS ONE